# Development and Evaluation of a New Effective Tool and Method for Assessing *Varroa*
*destructor* (Acari: Varroidae) Mite Populations in Honey Bee Colonies

**DOI:** 10.3390/insects13050457

**Published:** 2022-05-12

**Authors:** Francisco J. Posada-Flórez, Samuel K. Abban, I. Barton Smith, Steven C. Cook

**Affiliations:** USDA-Agricultural Research Service, Bee Research Laboratory, 10300 Baltimore Avenue, BARC-East, Beltsville, MD 20705, USA; francisco.posada@usda.gov (F.J.P.-F.); samuel.abban@usda.gov (S.K.A.); smithib731@gmail.com (I.B.S.J.)

**Keywords:** *Apis mellifera*, mites, PVC, sampling, sugar roll

## Abstract

**Simple Summary:**

To avoid losses of honey bee colonies, beekeepers should monitor the populations of the harmful ectoparasitic mite, *Varroa destructor* (Anderson–Truman). Here, we describe a novel device and method for monitoring mite populations. The Varroa Shaker Device (VSD) is constructed of sections of polyvinyl chloride (PVC) pipe that utilizes screens of two mesh sizes that separate the bees from the mites and captures the mites for easy counting. The VSD can be used by shaking bees with only water as the wash solution but may also be used with alcohol wash solutions; the VSD accurately assessed mite loads using fewer than 250 bees and shaken with 250 mL of water for one minute. The recovery of mites using the VSD is >90%, which is such as that recorded for using the commonly used sugar roll method but is easier to use in both laboratory and field settings. Additionally, tests demonstrated that for the VSD to assure accurate mite counts are achieved, honey bees should be taken from frames with an open and/or capped brood where the mites are more likely located.

**Abstract:**

A new device for assessing *Varroa destructor* (Anderson–Truman) mite infestations in honey bee colonies was designed, tested, and evaluated against the sugar roll method, a widely used method by beekeepers. The Varroa Shaker Device (VSD) is constructed of polyvinyl chloride (PVC) pipe that separates into three parts. Inside the shaker there are two mesh sizes; the larger mesh separates the bees from the mites, and the smaller mesh captures the mites. The VSD can be used by shaking bees with only water as the wash solution. The recovery of mites using the VSD is >90%, which is such as that recorded for using the sugar roll method. Our tests demonstrated that the VSD accurately assessed mite loads when fewer than 250 bees were sampled and shaken with 250 mL of water for one minute. To assure accurate mite counts are achieved with any sampling device, honey bees should be taken from frames with an open and/or capped brood where the mites are more likely located. The VSD can be used in both laboratory and field settings to accurately assess honey bee colonies for levels of mite infestation or for collecting live mites for research purposes.

## 1. Introduction

*Varroa destructor* (Anderson and Truman, 2000 (Parasitiformes: Mesostigmata: Varroidae) (hereafter, Varroa)) is an ectoparasitic mite of honey bees, and is considered the most economically important pest of *Apis mellifera* (L.) colonies worldwide. Colony losses stemming from infestations of these mites remain an ongoing threat for the U.S. beekeeping industry. Varroa cause direct damage from feeding on adult and immature honey bees, and vector pathogenic viruses that decrease the life span and performance of the infected bees [1]. Intervention by the beekeeper is required to guarantee the survival of honey bee colonies infested with Varroa. The best approach to control these harmful mites is to use an integrated pest management strategy [2], which consists of using different chemical and non-chemical control methods, the success of which depends on making an accurate assessment of the Varroa infestation levels. Remedial actions against mites are suggested when mites are above a threshold of ~3–5 mites/100 bees [3].

There are a number of methods for gauging Varroa populations in honey bee colonies that have been used in a variety of studies requiring accurate measures of mite populations [2,4,5,6]. The best method to assess Varroa infestations should be rapid, allowing quick decisions by beekeepers on whether to apply treatments, and based on sampling methods that accurately depict the standing population of mites in the colony [3,7,8,9]. Basic methods to assess Varroa populations include visual inspections of brood frames and checking nurse bees for mites physically attached to their bodies. Using this method, a beekeeper can detect the presence of Varroa, but cannot easily estimate the mite population based on the number of mites per 100 bees, for example. The most accepted methods for gauging mite populations in honey bee colonies include installing sticky boards, and collecting bees and performing either a sugar or ether roll, and/or alcohol wash [4,10,11,12].

Sticky boards, which are installed on the floor of the hive, passively trap mites fallen from hive substrates or bees above, whether from the grooming behaviors of the bees dislodging the mites or losing grip on the bees. Additionally, mites trapped on a sticky board may have fallen when they die naturally, or when dead mites are removed from cells by hygienic worker bees. Although natural mite drop can be a good indicator of whole-colony mite loads [9,13], making accurate counts of mites on sticky boards can be difficult—scanning a sticky board under magnification can be tedious, and mites can be mistaken for debris, or the position of a fallen mite on the sticky board may make it difficult to observe. Finally, without some measure of the bee population it is difficult to calculate the standard measure for the number of mites per 100 bees used to determine whether treatments should be applied.

The sugar roll method uses fine-powder confectioners sugar, which coats the collected bees and helps to dislodge their adherent mites during shaking in a screened vessel (usually a wide-mouth, 1 pint (~568 mL) Mason jar) [4]. Shaking time may vary, and mites fallen through the screened lid during shaking fall into a shallow pan filled with water. Then, the water can be sieved, and mites collected, or mites floating on the surface of the water may be counted. Many mites survive this step, making this method useful if living mites are required. This method is widely accepted [14], but in practice, especially when used in a field setting, does have some limitations. First, the powdered sugar is highly hydroscopic, thus when the relative humidity is high, the sugar can stick to the sampling device and/or the bees, making it more likely Varroa may remain attached to the bees or be heavily coated with sugar. Second, an alarm pheromone released by captured bees during shaking may attract bees from surrounding colonies, making it difficult to sample many hives during one outing. Another inconvenience of the sugar roll test is that the sugar dust generated during shaking may become a health hazard, requiring the user to wear a mask. Finally, an accurate count of shaken bees is difficult to assess with this method.

Instead of using confectioners’ sugar, bees collected into a lidded vessel can be soaked with either ether or alcohol (isopropanol or ethanol). Then, after vigorous shaking, sometimes with a mechanical rotary shaker, the wash solution is sieved and any mites collected and their numbers are recorded [4,5,12]. However, using an ether/alcohol wash exposes the user and environment to potentially hazardous wastes. Moreover, the ether/alcohol poses a fire risk if it comes into contact with a lit smoker in the field [5,11]. As with the sugar roll method, if it is required to be as accurate as possible, the user may repeat washes if necessary. Using ether/alcohol as the wash solution kills the bees and the mites.

Users considering which method to use for gauging Varroa populations should also consider other important factors for obtaining accurate estimates, including the number of bees to collect, from where in the hive the bees are collected, and the time needed to shake the sample. First, some beekeepers recommend sampling a few bees to more than 300 bees per sample. Removing large numbers of bees for making mite counts may irreparably harm small honey bee colonies, but too few (e.g., <100) will not allow the calculation of the standard threshold measure for the number of mites per 100 bees. The second factor to consider is where from within the colony to take the bees for sampling. For example, Varroa show preference for nurse-aged bees over foragers [15], and bees collected from frames of honey can yield different numbers of mites (usually lower) than bees collected from brood frames [10]. Finally, the time spent shaking the vessels containing the samples, whether in sugar or ether/alcohol, can impact the efficacy of complete removal of all mites from collected bees. This final point raises another issue related to having a standard type of shaking vessel for making estimates of Varroa populations [16]. To this end, we have designed and constructed a Varroa Shaking Device (VSD), which has a low build cost, and provides quick and accurate estimations of mite populations, while posing a minimal hazard to the beekeeper. We assessed the efficiency of the VSD to dislodge mites from different sample sizes of bees, and determined the optimal time required to shake the VSD to dislodge the mites from bees. We also re-evaluated the location of the bees in the hive that allow for the best estimation of Varroa infestations, and finally we compared the efficiency of the VSD versus the sugar roll method to dislodge mites from the bees.

## 2. Materials and Methods

### 2.1. Construction of the Varroa Shaking Device (VSD)

The VSD consists of a segment of polyvinyl chloride (PVC) piping enclosing two types of mesh to retain both the bees and to filter and collect the Varroa (Figure 1). The advantage of using this device over others is that the mites are separated out from the bees and retained on a screen that facilitates easy counting. The main segment of tube is 15 cm long and 5.8 cm wide (6.0 × 2.0 inches). On both ends of the tube are threaded male–female adaptors. One end is used to load the bees, which has a top cap to close the shaker and avoid any bees escaping. A radiator funnel (*e.g.*, FloTool model #10703) may be used to facilitate the addition of bees into the device. On the other end of the tube, before the threaded female–male adaptor is attached, a piece of metal mesh (#8, 5.0 mm mesh) is glued (PVC glue) to the distal end that retains the bees and filters the mites that are dislodged from the bees. Additionally, another piece of metal mesh (#80, 0.5 mm mesh) is glued to the distal end of the male attachment that connects to the female thread to capture the mites. After this attachment, a cap is attached to close the shaker device (Figure 1). The device is inverted and the wash solution, in this case 250 mL of water, is poured inside through the larger mesh screen at the bottom. Next, the fine mesh screen portion of the VSD is screwed onto the device, and finally, the base cap is attached before again inverting to the original orientation. The VSD is now ready to be shaken for dislodging the mites from the honey bees. The VSD is operated by gently shaking the device in an up and down motion within a set time interval. After shaking, and holding the device vertically, the top lid is loosened slightly to release pressure, making it easier to remove the bottom cap. Removal of the bottom cap releases the wash solution, thus the user should open it over a collection basin if wash solution is to be saved or reused. Once open, mites are retrieved by removing the portion with the fine mesh screen, which captured the mites for easy collection. At this point, the Varroa can be removed from the screen and placed on a tray lined with paper towels for counting. To capture any remaining mites not dislodged during shaking, the fine mesh screen portion may be reattached, and then water run through the open top for 1 min. Very infrequently we noted that mites would be lodged in the crevice where the mesh screens are glued onto the lower section of the device. A fine paintbrush can be used to retrieve these mites.

**Figure 1 insects-13-00457-f001:**
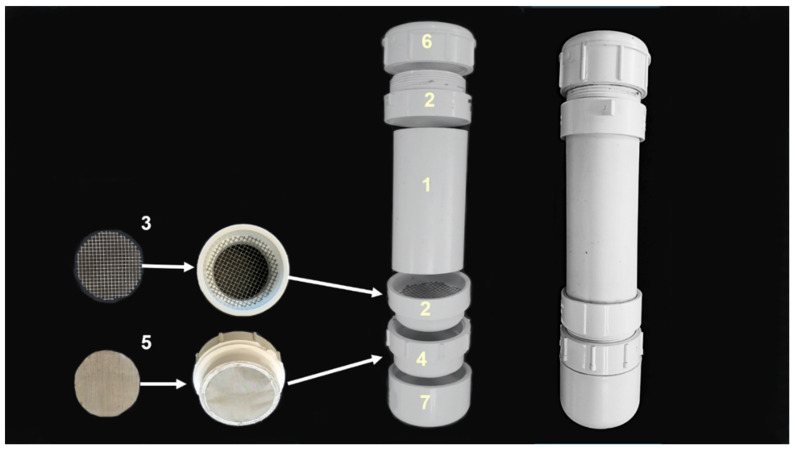
Schematic of the components comprising a Varroa Shaker Device (VSD) and how to assemble the components. The numbers correspond to the item numbers listed in Table 1.

### 2.2. Evaluations for Using the VSD

#### 2.2.1. Test of Sample Size (Number of Bees)

To evaluate whether the number of bees comprising the sample affects the efficiency of the VSD to dislodge Varroa, frames containing open and partially capped brood or emerging workers were taken from each of 47 honey bee colonies located on the grounds of the USDA-ARS Beltsville Agricultural Research Center, Beltsville MD (all subsequent experiments described herein used colonies from this location). The bees from each frame were shaken off into a shallow plastic pan. Samples of bees were collected by using a 1/3 cup (~78 mL) scoop to collect bees into individual glass jars labeled with the colony ID. Experience demonstrated that this volume of bees translated to approximately between 200 and 300 bees, which is the number of bees suggested by others for sampling bees for monitoring Varroa [7,17,18,19]. The samples of bees were taken without measuring, with the aim that the number of bees in the scoop were somewhat variable, as this estimates how the number of bees sampled affects the recovery of mites. The bees were taken from jars and placed in separate, labeled devices, then 250 mL tap water was added as previously described. The VSD was shaken for one minute, and after collecting mites trapped on fine mesh screen, tap water was run through the device for one minute. The above process was repeated once more after the first collection of mites were removed. When the process was completed, the bees were removed from the VSD and placed separately on a tray with water for meticulous visual examination for any Varroa that remained in the sample attached to the bees. We also counted the number of bees in each sample.

After determining the number of bees comprising each sample, they were separated into two groups having either <250 or >250 bees (no samples were comprised of exactly 250 bees), and the efficiency of mite recovery (i.e., percent recovery of total mites) was determined for each group of samples. We estimated the efficiency of the VSD for recovering mites using the equation E = ((N_1_ + N_2_)/(N_1_ + N_2_ + N_3_))*100, where N_1_ equals the number of mites recovered from the first shaking, N_2_ equals the number of mites recovered from the post-shaking rinse, and N_3_ equals the number of mites recovered from the final examination of bees. The efficiency of mite recovery was compared between the two sample size groups using a non-parametric Kruskal–Wallis test [20].

#### 2.2.2. Test of Device Shaking Time

Tests were conducted to determine the time required to shake the VSD for optimally efficient mite recovery using water as the wash solution. For this, approximately 250 honey bees were collected from multiple brood frames (as above) to ensure the presence of Varroa in collected sample. The samples were then placed in the VSD and shaken by hand for 40, 50, 60 or 90 seconds. Once opened, and after removing and counting mites on the small mesh screen, the VSD was then washed under tap water for one minute over a fine mesh sieve to capture any mites that were not removed from shaking. The process was repeated one more time to recover any remaining mites. After the second shake and rinse, the bees were removed from the VSD and then placed on a tray with water and meticulously visually examined for any remaining mites. The efficiency of mite recovery (as percent of total mites removed) was compared between the four shaking periods using a non-parametric Kruskal–Wallis test for multiple comparisons (each pair) [20].

#### 2.2.3. Test of Sampling Location within Hives

The location inside colonies where honey bees are sampled for gauging mite populations may determine the accuracy of mite counts in the colony. Thus, tests were conducted that gauged the mite infestation rate of colonies when bees were collected from different locations within the hive: (1) from frames with open and partially capped brood or emerging workers; (2) from frames with open honey and pollen; and (3) a blend from both the first two sampling sites. This sampling regime was intentional with the goal of sampling nurse bees, forager and/or food processing bees, and a mix of these individuals, respectively. The test was conducted using six replicate colonies. For each colony, bees collected from each sampling site were placed in a large shallow pan, then approximately 250 bees were scooped out and placed into a VSD. Care was taken to collect a similar amount (volume) of bees in each scoop taken from each of the two sampling locations. The bees were shaken, and mites collected as described above. The total numbers of mites and bees comprising each sample were compared across sampling location using ANOVA with Student’s post hoc tests. To determine whether the number of mites removed from bees depends on the location within the hive bees are collected, and thus giving a better representation of colony mite loads, the number of mites per 100 honey bees was analyzed using a non-parametric Kruskal–Wallis test for multiple comparisons (each pair) [20].

#### 2.2.4. Evaluation of the VSD Compared with the Sugar Roll Method

Evaluations were conducted in early September of both 2017 and 2018. For the 2017 and 2018 tests, nurse bees were collected from a frame containing open and partially capped brood or emerging workers from each of 21 and 10 honey bee colonies, respectively. The bees from each frame were shaken off onto a shallow plastic tray, and after gently homogenizing bees together, two samples of bees were collected per colony by using a 1/3 cup scoop and placed in two glass pint Mason jars labeled with its respective colony ID, one to test the sugar roll method and the other to test the VSD.

The bees collected for the sugar roll method were shaken as recommended by [11,18,19,21,22]. Thus, two tablespoons of powdered confectioners sugar (Domino Foods Inc., Yonkers, NY, USA) was added to the jar with the bees, and the jar was rolled ~one minute to assure an even coating of the bees with sugar. After the initial roll, the jars sat for three minutes to allow bees to groom off the mites. To dislodge Varroa from bees using the VSD, water was added to the device and then shaken for one minute. Afterward, the bottom cap and screened section holding mites was removed from the VSD, and water was continuously poured through the device for one minute over a sieve to catch any remaining mites. The same samples were re-evaluated once again using the VSD and sugar roll methods to compare the efficiency of methods. When the sugar roll and the VSD process was completed, the bees from each sample were separately placed on a tray with water for examination and meticulous counting of any mites that remained attached to the bees. The number of bees in each sample were also recorded. The number of mites and bees comprising each collected sample, and these data represented as the number of mites per 100 sampled bees, were compared between years using a non-parametric Chi-Square Kruskal–Wallis test for multiple comparisons (each pair) [20] to determine difference between years prior to testing the efficiency to remove mites from collected bees using the different sampling methods. Depending on whether there were differences between evaluation years in any of the above variables, the efficiency of mite removal (percent removed) using either the VSD or sugar roll methods was evaluated for each year separately or combined using a non-parametric Chi-Square Kruskal–Wallis test. Data for the first and second shaking periods, as well as the combined data, were tested separately.

## 3. Results

### 3.1. Test of Sample Size (Number of Bees)

The total number of bees and Varroa comprising all samples for this test were 11,209 and 1338, respectively. The average number of honey bees comprising the samples, the number of mites recovered from these samples, and the number of mites/100 bees are provided in Table 2. The number of bees for the low and high sample sizes ranged between 128 and 247 bees and 253 and 407 bees, respectively. For the first shaking episode, the percent mite recovery was not significantly different between sample sizes (*X^2^* = 1.2533, df = 1; *p* = 0.2629) (Figure 2). The second shaking episode of samples increased the efficiency of mite recovery (Figure 2), and as above, these values were not significantly different (*X^2^* = 0.9043, df = 1; *p* = 0.3416). The final careful visual examination of the samples having < 250 and >250 bees yielded an additional 3.8 ± 2.6 and 6.9 ± 3.6 percent, respectively, of the total mites remaining (Figure 2). The efficiency of mite recovery following the final inspection of bees for mites did not differ between samples sizes (*X^2^* = 0.0035, df = 1; *p* = 0.9529).

### 3.2. Test of Device Shaking Time

The total number of bees and Varroa comprising all samples for this test were 3775 and 370, respectively. The average number of honey bees comprising the samples, the number of mites recovered from these samples, and the number of mites/100 bees are provided in Table 3. The average level of mite infestation ranged between 6.4 ± 5.4 and 10.0 ± 9.8 mites per 100 bees. Across the 40, 50, 60, and 90 second shaking times, the first shaking removed over 80% of the Varroa present in the bee samples. For the first shaking episode, the efficiency of mite recovery increased with longer shake times; however, there was no significant difference in mite recovery among the four shake times evaluated (*X^2^* = 2.98; df = 3; *p* = 0.3945) (Figure 3). For the second shaking episode, the highest mite recovery was achieved at 40 seconds and the lowest recovery was for 90 s (Figure 3); however, there was no significant difference in mite recovery between the four shake times evaluated (*X^2^* = 2.98; df = 3; *p* = 0.3956). The final examination of bees for any remaining mites showed only the 60 second shake resulted in any additional mites not dislodged by the two shaking episodes (Figure 3).

### 3.3. Test of Sampling Location within Hives

The total number of bees and Varroa comprising all samples for this test were 6876 and 121, respectively. The average number of honey bees comprising the samples, and the number of mites collected from the samples collected from each location within hives are provided in Table 4. The number of bees and Varroa comprising each sample were not significantly different (*X^2^* = 4.01; df = 2; *p* = 0.1345) and (*X^2^* = 4.37; df = 2; *p* = 0.1127), respectively, but with samples collected from brood frames having marginally non-significantly higher number of mites than those collected from forager frames (*p* = 0.0632) (Table 4). The level of mite infestation measured as the number of mites/100 bees was not significantly different between sampling locations (*X^2^* = 4.45; df = 2; *p* = 0.1083). One colony had a much lower mite infestation (2.4 mites/100 bees) in the sample collected from brood frames compared to a mean of 11.4 ± 2.2 mites/100 bees from samples collected from the remaining five colonies at that location. Removal of this one data point prior to running the ANOVA resulted in a significantly higher mite infestation level (mites/100 bees) in samples collected from brood frames than those collected from either forager frames or a blend of the two sources (*X^2^* = 6.96; df = 2; *p* = 0.0308) (*F*_2,14_ = 4.93, *p* = 0.0240).

### 3.4. Evaluations of the VSD Compared with the Sugar Roll Method

All evaluations in both 2017 and 2018 were conducted using bees sampled from frames having open and partially capped brood or emerging workers. Summary statistics for the number of bees and Varroa comprising the samples, as well as these data presented as the number of mites per 100 collected bees, are given in Table 5. There were no significant differences between years or method from *post hoc* tests in these variables.

Given there were no significant differences in the sampling parameters for each evaluation year, the data for the efficiency of Varroa removal (percent mites recovered) were combined prior to running analyses for testing between sampling methods. For the first shaking period, the VSD and sugar roll methods recovered 81.1 ± 3.8 percent and 96.0 ± 3.7 percent of mites in samples, respectively, and with these data the sugar roll method recovered significantly more mites (*X^2^* = 7.35; df = 1; *p* = 0.0067) (Figure 4). For the second shaking period, the VSD and sugar roll methods recovered 9.4 ± 2.1 percent and 1.9 ± 2.1 percent of mites, respectively, and with these data the VSD recovered significantly more mites (*X^2^* = 4.45; df = 1; *p* = 0.0348). For the first and second washes combined, the VSD and sugar roll methods recovered a total of 90.6 ± 2.2 percent and 97.9 ± 2.2 percent of mites, and with these data the sugar roll method recovered significantly more mites overall than the VSD (*X^2^* = 4.09; df = 1; *p* = 0.0431).

## 4. Discussion

The VSD is a simple, yet solid tool that is easy to handle, and when using water as the wash solution, is safer to operate in both laboratory and field settings compared to methods using confectioners’ sugar or alcohol to dislodge mites from bees, thereby saving time and resources. For both field and laboratory use, the VSD has several advantages over using other sampling methods to determine mite infestation levels in honey bee colonies. First, fewer items are necessary to prepare the VSD for use than the sugar roll method, which requires items, such as jars, sugar, water, and a strainer, for its use. Second, using the VSD does not pose the same physical problems to accurately measure mite infestation levels as the sugar roll method; the sugar, being highly hydroscopic, can cause mites stick to the surfaces of the glass jars, or to the bees, especially in conditions having high relative humidity. Additionally, if the mites collected must remain alive for subsequent research, the VSD using water allows up to 80% of the mites to survive (authors’ unpublished data); this is similar to mite survivorship when using the sugar roll method [23]. For practical work, we recommend 60 second shaking intervals of the VSD because it is a reasonable time to process and evaluate a sample, and the recovery rate using a single one-minute shaking interval was sufficient to recover adequate numbers of mites. Another factor to consider is that longer shaking times may result in more debris being trapped with mites on the fine screen.

The best estimation of Varroa infestation can be made when bees are sampled from frames with many open and partially capped brood cells or emerging workers, where female Varroa are in transit and ready to move into brood cells. Underestimation of Varroa infestation levels can be made when the bees are taken from places where no brood is present, or brood is completely capped. A possible reason as to why less Varroa are found in these scenarios is that they do not prefer parasitizing bees that are engaged in activities in which they would have less chance of entering brood cells to continue their life cycle (e.g., forager-aged bees) [24]. Another possible reason is that Varroa prefer nurse bees because these bees present the best possible chance for them to transition into open brood cells, which is less likely if mites are attached to older bees that have functional Nasonov glands that release a volatile pheromone that repel mites [25,26].

Overall, the evaluation of the VSD shows that it has similar performance compared to the sugar roll method, which is reported to be the best method to assess and recover Varroa [14]. This indicates that the choice between the two methods should be based on cost, ease of use, and any potential hazards for the user. There are some distinct differences between the two methods, particularly that the sampled bees are not killed from using the sugar roll method. However, the acceptance rate of bees by their home colonies, or their longevity after sugar shaking are not known. Both methods allow mites to survive if they are needed to run subsequent experiments; they need only to be placed in a container lined with a wet tissue until used for laboratory studies. The VSD method, however, does not coat the mites in powdered sugar, which may compromise their survival.

The evaluation also shows that to obtain the best performance from both shakers, one should consider the maximum number of bees that can fit into the shakers to dislodge mites and not become stuck on the other bees or in the sugar on the jars. The results show that a sample of around 250 bees, which is about a 1/3 measuring cup, is an appropriate number of bees for using the VSD, while for the sugar roll method, the recommendation is 300 bees [3,7,18,19,25,27,28], and for alcohol wash it is 100 bees [7]. To ensure an accurate estimation of Varroa infestation, sampled bees should be shaken twice. Using the VSD in this manner obtains an equivalent efficiency rate to that of the sugar roll method at the highest efficiency rate presented in this study. Additionally, the efficiency of the VSD to recover mites from samples of honey bees was evaluated using only water. However, the device can also work with most liquids other than water to dislodge mites, except for ether, which can react with the PVC material. Finally, it can be pointed out that the VSD can simplify the assessing and evaluation of the Varroa infestation. This device can work in all settings—laboratory or in the field—and the main advantage is that it is inexpensive to build and uses water, which is safer and cheaper than other solvents.

## Figures and Tables

**Figure 2 insects-13-00457-f002:**
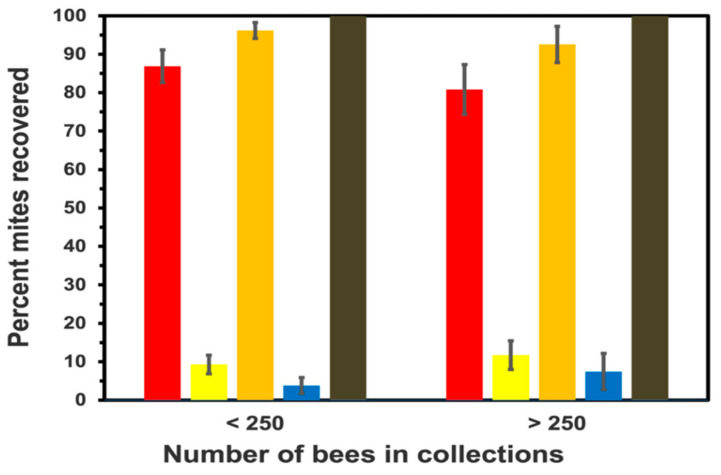
The percentage (mean ± standard err (se)) of Varroa dislodged from fewer or more than 250 collected honey bees after shaking the Varroa Shaking Device (VSD) for one minute with water (<250 N = 31; >251 N = 16 trials). The efficiency was estimated for first (red bar) and second (yellow bar) shaking periods, followed by the residual percentage collected from careful inspection of bees after the second shaking period (blue bars). The sum of percent recovery between the first and second shaking period and the second shake period and final visual inspection of bees for residual mites, are given by orange and black bars, respectively.

**Figure 3 insects-13-00457-f003:**
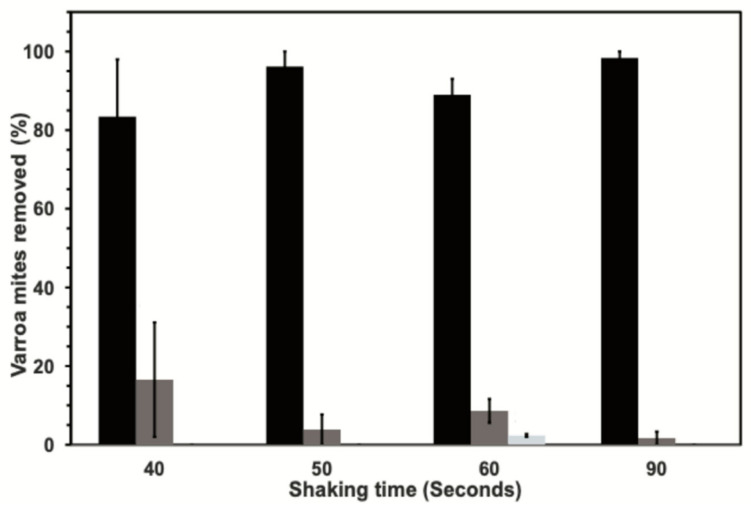
The percentage (mean ± standard err (se)) of Varroa dislodged from collected honey bees by using four different shaking periods. The percentage of mites recovered from the first and second shaking periods are given by black and dark-gray bars, respectively. The percentage of residual mites collected from final visual inspection of bees is also given where observed (light-gray bars).

**Figure 4 insects-13-00457-f004:**
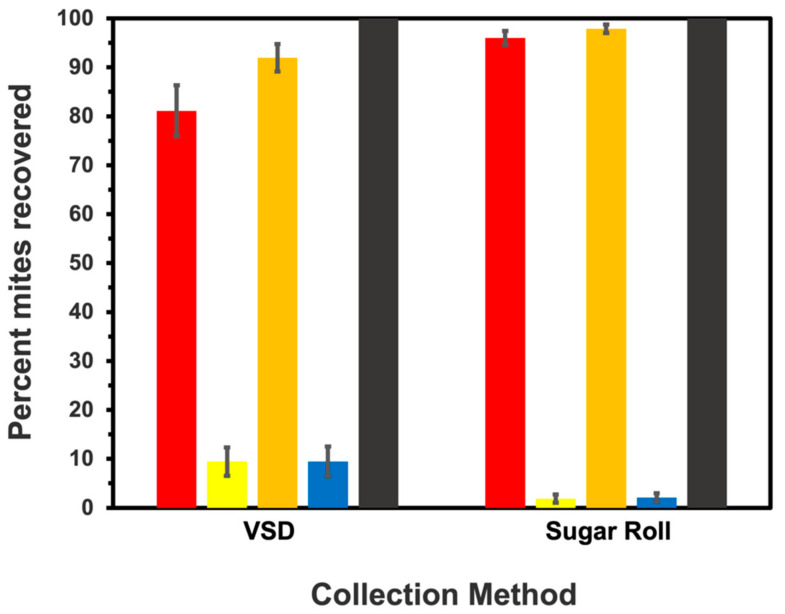
The percentage (mean ± standard err (se)) of Varroa dislodged from honey bee samples using the Varroa Shaking Device (VSD) and the sugar roll method. The efficiency was estimated for first (red bar) and second (yellow bar) shaking periods, followed by the residual percentage collected from visual inspection of bees after the second shaking period (blue bars). The sum of percent recovery between the first and second shaking period and the second shake period and final visual inspection of bees for residual mites are given by orange and black bars, respectively.

**Table 1 insects-13-00457-t001:** List and descriptions of materials needed for building a Varroa Shaker Device (VSD). All polyvinyl chloride (PVC) components are schedule 40 (i.e., ~0.4 cm wall thickness). All width (W) measurements given are for the inner diameter; L = length of component.

Item No.	Material	Description	QTY
1	PVC * Pipe	Main Body, 5.8 cm W, 15 cm L	1
2	PVC Adaptor	Threaded Male–Female, 5.8 cm–6.0 cm W, ~4.8 cm L	2
3	Metal Mesh	#8 Wire Mesh 6.0 cm W	1
4	PVC Spigot	Threaded Female–Male 6.0 cm–5.8 cm W, ~4.8 cm L	1
5	Metal Screen	#80 Wire Mesh 5.8 cm W	1
6	PVC Top Cap	Threaded, 5.8 cm W, ~3.5 cm L	1
7	PVC Bottom Cap	Non-threaded, 6.0 cm W	1

* Polyvinyl chloride.

**Table 2 insects-13-00457-t002:** Summary statistics for tests of sample size (number of bees) on estimation of Varroa recovery efficiency. The numbers (mean ± standard err (se)) of Varroa collected in each of the bee samples, the number of bees in each sample, and number of mites adjusted as the number of mites per 100 honey bees.

Sample Size Group	No. Hives	No. Varroa	No. Bees	No. Varroa/100 Bees
<250 bees	31	18.5 ± 2.7	218.0 ± 4.2	8.4 ± 1.2
>250 bees	16	47.7 ± 25.9	278.2 ± 11.6	17.9 ± 9.8

**Table 3 insects-13-00457-t003:** Summary statistics for tests of VSD shaking time for estimation of Varroa recovery efficiency. The numbers (mean ± standard err (se)) of Varroa collected in each of the bee samples, the number of bees in each sample, and number of mites adjusted as the number of mites per 100 honey bees.

Sample Shaking Time (Seconds)	No. Hives	No. Varroa	No. Bees	No. Varroa/100 Bees
40	4	19.0 ± 9.8	243.8 ± 11.43	8.5 ± 4.6
50	3	16.7 ± 11.6	241.0 ± 19.7	6.3 ± 4.5
60	4	22.8 ± 8.5	229.8 ± 15.4	9.5 ± 3.2
90	4	38.3 ± 17.5	233.8 ± 11.8	15.5 ± 6.4

**Table 4 insects-13-00457-t004:** Summary statistics (mean ± standard err (se)) for the numbers of Varroa collected in each of the bee samples, the number of bees in each sample, and number of mites adjusted as the number of mites per 100 honey bees used in tests of location in hive where honey bee samples are collected for estimation of Varroa recovery efficiency.

Location Bees Collected	No. Hives	No. Varroa	No. Bees	No. Varroa/100 Bees
Brood Frame (BF)	6	21.5 ± 4.4	227.0 ± 10.5	9.9 ± 2.4
Forager Frame (FF)	6	9.8 ±2.6	218.8 ± 11.6	4.5 ± 1.1
Blend (BF + FF)	6	13.5 ± 2.6	241.8 ± 5.5	5.7 ± 1.2

**Table 5 insects-13-00457-t005:** Summary statistics (mean ± standard err (se)) for the comparison between the Varroa Shaker Device (VSD) and the sugar roll methods in their efficiency to remove mites from samples of honey bees for evaluations run in both 2017 and 2018.

Year and Method	No. Hives	No. Varroa in Sample	No. Bees in Sample	No. Varroa per 100 Bees	Removal Efficiency (%)
2017		Mean ± se	Mean ± se	Mean ± se	Mean ± se
VSD	21	20.2 ± 4.6	245.5 ± 12.9	8.3 ± 1.8	90.3 ± 4.2
Sugar Roll	21	22.5 ± 5.2	255.6 ± 12.6	9.1 ± 2.0	97.4 ± 1.3
2018					
VSD	10	38.4 ± 9.7	226.0 ± 3.70	17.0 ± 4.2	95.8 ± 1.3
Sugar Roll	10	34.5 ± 8.18	227.0 ± 5.27	15.0 ± 3.7	98.4 ± 0.7

## Data Availability

The data presented in this study are available on request from the corresponding author. The data are not publicly available due to patenting issues.

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
