# Peer review of "Development and Evaluation of a New Effective Tool and Method for Assessing *Varroa"

_insects, 2022, doi:10.3390/insects13050457_

Round 1
Reviewer 1 Report
This is a very pragmatic paper that provides a user-friendly tool for estimating the mite population in the field or laboratory, the varroa shaker device (VSD). The author is thoughtful and was able to solve the specific problems during the experiment operation. I think VSD is very useful for all the bee research scientists and bee-keepers to monitor the varroa population dynamic. From the data, VSD has the comparative efficiency in evaluating varroa level in hives though the recovered mites are lower than the sugar roll method, that does not reduce the advantage of the VSD. There are still some minor considerations that for the “test of sample size” experiment, the different sampled bee sizes come from different hives, is that properly to evaluate the size effector? Maybe the sampled bees with small size “< 250 bees” or bigger size “> 250 bees” come from the same hive should be more convictive.
L304, Table 2. The “A” letter in the table beside the number of each line may be indicating the statistic meaning “no significant difference” between groups. It would be better to state in the table legends.
Author Response
There are still some minor considerations that for the “test of sample size” experiment, the different sampled bee sizes come from different hives, is that properly to evaluate the size effector? Maybe the sampled bees with small size “< 250 bees” or bigger size “> 250 bees” come from the same hive should be more convictive.
Thank you for the comment: We feel that sampling from the same colony, without replacing the bees and mites removed from the first sampling may have affected the populations of both bees in mites in the colony and thus affected the outcome of the repeated sampling events for each colony. Thus, we feel it was better to take samples from different colonies.
L304, Table 2. The “A” letter in the table beside the number of each line may be indicating the statistic meaning “no significant difference” between groups. It would be better to state in the table legends.
Thank you for your comment: We have removed the letters from Table 2 as the same statistical results were presented in the paragraph from the results section.
Reviewer 2 Report
Title: Development and evaluation of a new effective tool and method for assessing Varroa destructor (Acari: Varroidae) mite populations in honey bee colonies
In this study, Posada-Flórez al. present a device to assess Varroa mite populations in honey bee colonies.
In my opinion, the work presents and justifies the background that gives rise to the realization of this new device. The work objective is well defined, the introduction is well-argued and referenced, the results are interpreted appropriately and the conclusion is well justified. The authors discuss the advantages and disadvantages of the device and value the interest of this new device.
The methodology is well explained, although in my opinion sometimes it is not clear how many replicas there are for each of the tests carried out. When various study groups are made, how many times is each one replicated? (for example, when analyzing the number of bees needed or the shaking time).
Sometimes, in my opinion, a table with the results is missing, where it is very quick to see the values obtained from bees and mites (such as table 2).
The discussion compares the results obtained against the other varroa infestation estimation system in hives (i.e. sugar roll method) and presents the advantages and disadvantages of the two systems.
Minor comments:
Line 9 and 37: different font sizes appear
remove parentheses before Anderson
No further comments
Author Response
The methodology is well explained, although in my opinion sometimes it is not clear how many replicas there are for each of the tests carried out. When various study groups are made, how many times is each one replicated? (for example, when analyzing the number of bees needed or the shaking time).
Thank you for the comment: We have included additional tables that give the sample sizes we used in the different experiments. Because the populations of mites and bees could be altered from repeated sampling, we used each colony only once for each of the different experiments (separated by some time). Thus, we did not replicate within each colony but rather replicated across colonies.
Sometimes, in my opinion, a table with the results is missing, where it is very quick to see the values obtained from bees and mites (such as table 2).
Thank you for the comment: We have added additional tables giving the summary statistics giving the mean and standard errors for the numbers of bees and mites that comprised each of the samples.
Line 9 and 37: different font sizes appear
Thank you. We have made the correction.
remove parentheses before Anderson
Thank you. We have made the correction.